# Social–Emotional Learning of 2-Year-Olds Within Peer Interactions in Early Childhood Education Settings: A Scoping Review

**DOI:** 10.3390/bs15101303

**Published:** 2025-09-24

**Authors:** Yuru Ji, Jayne White, Shweta Sharma

**Affiliations:** School of Social and Cultural Studies, University of Canterbury, Christchurch 8041, New Zealand; jayne.white@canterbury.ac.nz (J.W.); shweta.sharma@canterbury.ac.nz (S.S.)

**Keywords:** social–emotional learning, toddler, early childhood education, peer interaction

## Abstract

Research has highlighted social–emotional learning (SEL) as a critical domain for 2-year-olds (2YOs), as well as the crucial role of their peers in the Early Childhood Education (ECE) context. Although strong evidence indicates that children begin developing social–emotional skills before 24 months, limited research has explored SEL in 2YOs, particularly in the context of peer relationships. With global increases in the enrolment of 2YOs in ECE, 2YOs have engaged more frequently in complex peer group interactions that offer expanded opportunities for their SEL. This scoping review explores the existing literature on 2YOs’ peer-related SEL within ECE contexts in the last 10 years (2014–2025). The search reveals a predominant focus on adult–child relationships and home-based contexts. Consequently, 13 highly related empirical studies were identified from an initial corpus of 755 studies. Based on these 13 studies, this scoping review aims to map the following: (i) how SEL is defined for 2YOs, (ii) the methodologies and theoretical perspectives adopted to investigate 2YOs’ peer interactions, and (iii) the role of peers in 2YOs’ SEL. The findings indicate that most studies adopt quantitative methodologies grounded in developmental psychology perspectives, often relying on predetermined SEL indicators to assess and interpret 2YOs’ SEL and peer interactions. These findings underscore the need for more qualitative and in-depth investigations of 2YOs’ SEL in peer contexts. Additionally, there is a call for more diverse methodologies and study designs to deepen our understanding of this important area of early childhood development.

## 1. Introduction

The existing studies reflect the growing interest in the social–emotional learning (SEL) of 2-year-olds (2YOs) in ECE contexts, driven in part by the increasing global enrollment of 2YOs in ECE ([12]; e.g., [22]; [23]; [30]). Additionally, the age paradox ([33]) arises between two approaches: a discrete age-oriented curriculum and a generic approach that transcends age parameters. This age paradox results in varied peer experiences and differing SEL outcomes for 2YOs. Research has exposed the importance of 2YOs’ SEL across multiple domains and for long-term development ([14]; [50]; [59]). In this review, 2YOs’ SEL is conceptualised as a broad term encompassing their interpersonal and intrapersonal development within peer interactions. However, research on 2YOs’ SEL within peer interactions remains limited, partly due to prevailing assumptions that 2YOs may not benefit from peer interaction to the same extent as older children ([27]; [50]; [59]). Meanwhile, dominant research approaches tend to adopt a positivist approach, which relies on designed experiments and standardised measurements to examine the SEL of 2YOs, such as the interventional study of [37] ([37]), and the utilisation of pre-determined checklists in the studies of [38] ([38]) and [31] ([31]). While these have provided crucial insights through large-scale exploration, they often interpret 2YOs’ SEL as a universal and static construct without considering the nuanced manifestations and strategic orientations in their context-based social discourses. As such, this scoping review aims to contribute to the understanding of 2YOs’ SEL within their peer interactions in the context of ECE and inform potential directions for future research.

According to [45] ([45]), the origins of SEL are rooted in Plato’s philosophical thoughts with the goal of whole-child development ([45]), which advocates fostering all aspects of children’s growth, including the cognitive, social, emotional, and physical. Thereafter, the exploration of children’s SEL has gradually expanded to encompass multiple dimensions of child development, including psychology and sociopsychology, often with an underlying economic focus ([4]; [26]). In 1994, the term SEL was first proposed at the first conference of the Collaborative for Academic, Social, and Emotional Learning (CASEL), which indicated an overall understanding of social–emotional (SE) skills and competencies for development. Thereafter, it gradually became an overt approach to inform and guide teaching and learning for young children regarding their self-awareness, self-management, social-awareness, responsible decision-making, and relationship skills ([16]). Generally, most of the research focusing on 2YOs’ SEL defines it as the acquisition of a series of social–emotional skills in relation to the self and social competency and establishing relationships with others in social practices ([44]; [75]). However, several other studies focus on the 2YOs’ own social environmental construction and meaning-making within the interactions with others and the whole environment ([10]; [43]; [46]; [57]).

Although research has emphasised 2YOs’ social practices in the ECE setting, less research focuses on this age level generally, and much less is known about their SEL within peer interactions. The gap can be attributed to two key reasons: First, an emphasis on children’s cognitive development and school readiness has shifted the research focus toward older children in ECEC settings. The highlight of cognition leads to the focus on older learners in research rather than 2YOs or younger ([27]). For example, impacted by children’s development, it is assumed that peer influence becomes more significant once children reach preschool age, resulting in the perception that 2YOs are less socially responsive to peer interactions ([27]). Additionally, a strong connection between SEL and children’s developmental levels has reinforced this perception ([19]). This may cause the overlooking of the embodied social discourse and in-depth manifestations within the peer interactions of 2YO learners ([41]; [43]; [68]). ECE contexts have been suggested to expose 2YOs to rich and complex social situations and provide them with scenes to practice social–emotional (SE) skills ([46]). However, this developmental focus tends to view peers as significant others to reflect 2YOs’ SEL challenges ([47]; [75]) or facilitate 2YOs’ SEL, providing them with the Zone of Proximal Development (ZPD) ([74]). In this manner, 2YOs appear to take a relatively passive position in their social lives, while 2YOs’ subjectivity has been suggested to be formed through co-being with others within interactions through multi-mode communications ([55]; [69]).

Additionally, the prevalent caregiving-oriented positioning of 2YOs in educational practice results in insufficient research and interventions targeting their SE development. Consequently, the peers’ influence on 2YOs’ SEL is always overshadowed by external factors, such as adult facilitation and environmental structures ([52]; [60]). Evidence shows that the peer grouping of 2YOs in the ECEC contexts is frequently shaped by policies, state funding, and pedagogy constraints, including the teacher–child ratio, class management needs, and national curricula ([2]; [68]), rather than being guided by an understanding of their unique socialisation needs and peer groupings for their SEL ([72]). As a result, 2YOs often occupy a liminal position within ECE contexts, transitioning across varied peer settings and play environments ([39]). However, studies have increasingly underscored the importance of group life with peers for 2YOs ([18]; [40]; [61]); and presented 2YOs’ active engagement in group activities ([11]; [34]; [54]). While teacher-led intervention and scaffolding in 2YOs’ SEL has traditionally centred on supporting SEL in the early years ([8]), recent research suggested that teachers may have limited influence in facilitating 2YOs’ interpersonal, social, and emotional skills ([40]), particularly in peer group dynamics ([64]). Thus, the emphasis on caregiving and adult facilitation in ECE continues to marginalise the peer dimension of 2YOs’ SEL and overlook their active agency in social experiences.

In sum, 2YOs occupy a unique developmental stage, marked by the rapid growth in empathy, emotion regulation, and emerging self-awareness ([14]). This stage is acknowledged not simply as the extension of infancy, but rather as a sensitive period for social development ([58]). As 2YOs increasingly engage in group settings within ECE contexts, peer interactions become crucial for 2YOs’ SEL ([11]; [34]; [54]). Overlooking this age group risks missing essential opportunities for intervention and support, which may have long-term consequences for children’s social–emotional and mental health outcomes. In response, this scoping review seeks to address this gap and deepen the understanding within the field. Specifically, it explores how 2YOs’ SEL and the role of peers are conceptualised, as well as the methodology and theoretical frameworks adopted in the current study. Accordingly, three research questions are outlined below and will be addressed in the Results Chapter.

How is the SEL of 2YOs defined in the context of ECE?What methodologies and theoretical frameworks were utilised to investigate 2YOs’ SEL in ECE contexts?How does the current research inform our understanding of the role of peers in 2YOs’ SEL in the ECE context?

## 2. Materials and Methods

This study adheres to the scoping review framework proposed by [51] ([51]), which includes identifying the research questions, developing a review protocol, identifying relevant studies, selecting eligible studies, summarising the data, and reporting the results. Meanwhile, the review process and results are reported in accordance with the Preferred Reporting Items for Systematic Reviews and Meta-Analyses extension for Scoping Reviews ([53]) guidelines to ensure transparency and methodological rigour.

### 2.1. Information Sources and Search Strategies

To address our research questions, we conducted a wide-scope literature review searching eight databases, with the final search conducted in July 2025. They are Scopus, ERIC, Google Scholar, Education Source, JSTOR (education), and Taylor and Francis (including Journal of Early Childhood Education and Care, Early Education and Development, and European Early Childhood Education Research). These databases cover a wide range of relevant studies and also include field-specific sources to ensure the inclusion of significant research. During the keyword searching, Boolean operators are used to identify relevant literature, with the keywords emerging in all fields of the literature. The terms include “early childhood education”, “social–emotional learning”, “toddler”, and “peer interaction”, applying the operator AND to ensure relevant intersections between these keywords.

Given that the concept of SEL was introduced in 1994 (CASEL), the review considers SEL as a comprehensive framework encompassing related developmental dimensions. Several well-established frameworks and instruments, such as the Head Start Early Learning Outcome Framework ([32]) and Brief Infant–Toddler Social and Emotional Assessment ([5]) support the broad conceptualisation and global use of SEL in early years practice and research. Therefore, this review will take “social–emotional learning” as a unified term for searching, and paper identification, as well as the paper screening process. Additionally, the term ‘toddler’ is used to capture studies that specifically target or include two-year-olds as participants. ‘Peer interaction’ is employed to encompass both peer relationships and interactive processes relevant to social and emotional learning.

### 2.2. Eligibility Criteria

The inclusion and exclusion criteria were designed to ensure that the literature review aligned with the research objectives, demonstrated methodological rigour, and represented current knowledge in this field. Four specific inclusion criteria were applied. First, studies were required to include two-year-olds as part of the participant sample. Second, literature published in academic journals with full-text access available online was considered. Third, to capture both recent developments and foundational contributions in the field, the review included studies published in the last decade, between 2014 and 2025. This timeframe aligns with key developments in this field, such as UNESCO’s proposal of Sustainable Development Goal (SDG) 4.2 for ECE, which covers children aged from birth to 8 in 2015 and emphasises the play-based curriculum and peer engagement in ECE contexts ([63]). Similarly, the OECD expanded its focus on child for children from birth to school age in ECEC contexts in 2017, highlighting the importance of peer interactions for very young children’s development ([48]). Both initiatives underscore the significance of peer engagement in the SE development of very young children.

Lastly, the selected studies were located within certain fields of knowledge. To begin with, the review adopted “social–emotional learning” as the unifying concept, as SEL has been widely recognised as a holistic framework since its introduction in 1994 by CASEL. Therefore, studies that focus narrowly on specific educational subjects or special education, e.g., math, literacy, and art, without addressing broader SEL elements were excluded. To remain consistent with research objectives, in the full-text screening process, only studies based on ECE were included, which means that family/home-based studies were excluded. Additionally, to align with the review aim and its emphasis on educational perspectives on 2YOs’ SEL in the ECE settings, the selected studies were located in the field of education. Last but not least, studies focusing on child–teacher relationships were excluded in the first place, according to the study’s objective of exploring peer interactions. More details of the eligibility criteria are addressed in Table 1.

### 2.3. Study Selection

The study selection process followed the [53] ([53]) guidelines and was conducted in three phases. To maintain consistency, the three steps of paper selection were completed by a single author based on the established inclusion criteria. First, relevant literature was retrieved from eight selected databases using the inclusion criteria outlined in the previous section. In the second phase, titles and abstracts of the retrieved records were manually screened to assess their relevance to toddlers’ social–emotional learning. Studies were excluded at this stage if they focused solely on adult–child relationships, did not include two-year-olds as participants, targeted special education or specific subjects, or were conducted in home- or family-based settings. In the final phase, full-text articles were reviewed to confirm their eligibility for inclusion in the data extraction phase. Only empirical studies that met all inclusion criteria upon full-text screening were retained. A detailed overview of the study selection process is illustrated in the PRISMA-ScR flow diagram (see Figure 1).

### 2.4. Data Extraction and Charting

Data were systematically extracted using a deductive thematic approach from the 13 included studies according to the focus of each research question. The predefined themes are presented in the first row of each charting form. The data extraction process involved labelling the information in NVivo based on these predefined themes and summarising the results into structured forms. One author was responsible for extracting the data, while the other two authors reviewed the process and results. In case of any discrepancies, they collaboratively resolved the issues to ensure consistency in the final outcome. In Table 2, the information of selected studies’ characteristics, including the author(s) and year of publication, participant age, sample size, country, and study type is presented. The extracted data for further analysis is illustrated in a structured charting form under each research question in the Results section (Section 3).

As shown in the table, the 13 selected studies encompass a range of countries that have already incorporated 2YOs into their national ECE systems. However, the 13 studies are all situated within Western cultural contexts. Furthermore, three of these studies, published within the past two years, analysed secondary data, whereas the remaining earlier studies collected and examined primary data. Additionally, in terms of the age of participants, most studies include an expanded age range, instead of focusing specifically on 2YOs.

## 3. Results

This study aimed to examine how SEL in two-year-olds is conceptualised in relation to their peer interactions, and how the methodologies and theoretical frameworks used in the existing research inform our understanding of peer influence in early childhood ECE settings. A scoping review was conducted in order to map the current studies and identify the gaps in knowledge related to 2YOs’ SEL with peer interactions. The findings of the review are presented in relation to each research question in the following.

RQ 1: How is the SEL of 2YOs defined in the ECE context?

This research question explores how SEL in 2YOs is defined across the 13 selected studies and what the focus of each study is. Some essential information was examined and collected from each study, including the focus on SEL (i.e., interpersonal vs. intrapersonal relationships), the aspects of 2YOs’ SEL explored in the study, and the role of teachers within the research. Table 3 presents a summary of key information, with further discussion provided.

To begin with, the review revealed diverse conceptualisations of SEL, particularly in relation to peer interactions, reflected by the differences in study aims and focal areas. Of the 13 studies, nine emphasise the aspect of interpersonal relationships with peers in supporting 2YOs’ SEL, framing it primarily as learning that occurs through peer interactions, and also suggest the co-constructive environment within interactions. Among these nine studies, three defined SEL as encompassing both interpersonal competencies and the processes of individual development, situating SEL within a broad and holistic framework of child development. For example, [38] ([38]) focus on the 2YOs’ behaviours within peer play to assess not only 2YOs’ SEL, but also their gross motor development and well-being. Similarly, [27] ([27]) assessed multiple inter- and intrapersonal skills to examine how attendance in ECE settings impacts their SE development and school readiness. It suggested that centre-based ECEC at 24 months significantly influenced peer problems at 54 months. In addition, [62] ([62]) suggested a combined significance of these two parts by assessing both the biological and environmental factors that interplay with 2YOs’ SEL and skill development, with an emphasis on the use of diverse resources to foster 2YOs’ positive peer relationships.

The remaining six studies that focused on interpersonal aspects examined more specifically how peer relationships influence and are influenced by two-year-olds’ social–emotional learning and overall development. For example, [46] ([46]) and [20] ([20]) address 2YOs’ co-construction of the social meaning-making in SEL with their peers. In addition, [31] ([31]), [64] ([64]), and [65] ([65]) explored 2YOs’ interaction in peer play. [31] ([31]) suggests that social competence more strongly predicts participation in play than language skills. [64] ([64]) observed and assessed 2YOs’ social behaviour under a group-based approach. [65] ([65]) explored the 2YOs’ learning behaviours within peer interactions. Moreover, [42] ([42]) highlighted the interpersonal relationships in SEL, but with a focus on how teachers could build a quality environment for 2YOs’ peer interactions.

In contrast, the other four studies placed greater emphasis on cognitive development and the acquisition of social–emotional knowledge, positioning SEL as a developmental process that becomes evident through toddlers’ social behaviours and peer interactions. These studies foregrounded various aspects of intrapersonal development. For example, [37] ([37]) and [49] ([49]) highlight the emotional knowledge and development for SEL. [19] ([19]) focused on the role of social cognition and language development, highlighting the role of empathy and emotion recognition in navigating peer social environments, while [56] ([56]) offered a broader perspective on 2YOs’ intrapersonal development trajectories, including SE competencies and language development.

Regarding the role of teachers, among the 13 empirical studies included in this scoping review, 7 studies specifically highlighted the role of teachers in supporting 2YOs’ peer interactions and SEL through various approaches. Of these seven, four studies primarily emphasised the important roles of teachers in fostering intrapersonal development, such as social cognition, self-regulation, and emotional knowledge, including [37] ([37]), [49] ([49]), [19] ([19]), and [56] ([56]). These studies suggest that teachers’ interventions and guidance play a foundational role in shaping 2YOs’ peer interactions and SEL by supporting their internal capacities. Only one study, i.e., [42] ([42]), explicitly focused on interpersonal development, emphasising the significance of teachers in creating a high-quality social environment that facilitates peer relationships. This study positioned teachers as facilitators of children’s external social interactions. The remaining three studies adopted a dual focus, addressing both the interpersonal and intrapersonal aspects of SEL. These studies recognised the interplay between teacher support and children’s active participation in peer interactions, thereby promoting a more holistic view of 2YOs’ social development. For example, based on the teacher’s practices, [38] ([38]) assessed toddlers’ social–emotional behaviours during free play to understand their overall well-being and learning through peer engagement. Similarly, [27] ([27]) and [62] ([62]) explored how participation in early ECE, shaped by teacher practices and the quality of the learning environment, influences 2YOs’ SEL development.

The examination for the first question presents varied conceptualisations of 2YOs’ SEL in relation to peer interactions, shaped by differences in study aims and emphases. Nine studies focused on the interpersonal aspects of SEL, with three also considering intrapersonal development within a holistic framework. Four studies emphasised 2YOs’ cognitive and emotional processes, viewing 2YOs’ SEL as emerging through individual development. In addition, seven studies highlighted the role of teachers, with varying attention to intrapersonal, interpersonal, or combined influences. The next section explores how these studies approached peer interaction through different theoretical perspectives and methodologies, shaping how the role of peers was understood.

RQ 2: What methodologies and theoretical frameworks were utilised to investigate 2YOs’ SEL in ECE contexts?

The second research question examined the methodological approaches and theoretical frameworks employed by researchers to investigate two-year-old children’s SEL within the context of peer interactions across the selected studies. The analysis revealed a diverse range of research paradigms (i.e., qualitative, quantitative, or mixed), data collection methods, and theoretical orientations. This related information highlights the emerging trends in the adoption of research paradigms for studying 2YOs’ SEL within peer interactions in ECE settings. The specific methodological details and theoretical orientations are summarised in Table 4. Among them, studies by [46] ([46]) and [20] ([20]) adopted a pure qualitative paradigm with an interpretive approach, rather than using any established tools.

In terms of the paradigm, among the 13 studies, only 2 studies adopted a purely qualitative paradigm to explore and interpret 2YOs’ SEL within peer interactions. [46] ([46]) utilised video recordings to observe toddlers’ mealtime routines and analysed their peer interactions through discourse analysis, allowing for an in-depth examination of social dynamics in a naturalistic setting. Similarly, [20] ([20]) observed toddlers’ social play through video footage, focusing on their co-authored narratives and how these narratives related to the development of their identity of care. Both studies provided rich qualitative insights into the nuanced ways toddlers engage with peers in everyday contexts. Two additional studies adopted a mixed-methods approach, combining both quantitative and qualitative methodologies to explore SEL. [19] ([19]) used standardised tools to quantitatively assess 2YOs’ social behaviours, alongside teacher interviews to capture a more comprehensive interpretation of the children’s SEL, adding depth and context to the quantitative findings. Additionally, [64] ([64]) employed a mixed-methods approach, analysing video recordings of children’s peer interactions through a thematic qualitative analysis while also quantifying the frequency of specific social behaviours. These combined approaches provide both detailed narratives and measurable data on 2YOs’ SEL within peer interactions.

In contrast, the other nine studies adopted a purely quantitative approach to examine 2-year-olds’ SEL within peer interactions, utilising various rating and assessment tools targeting specific SEL aspects, as detailed in Table 4. For example, in the Norwegian context, [31] ([31]) and [38] ([38]) used parts of the Tras and Alle med observational materials to assess children’s development across domains such as language, well-being, play, social–emotional competence, and movement skills. Among these quantitative studies, six generated primary data, while the remaining three relied on secondary data, which are also longitudinal studies. [27] ([27]) using data from the Growing Up in New Zealand longitudinal cohort, explored the relationship between weekly ECEC attendance at 24 months and behavioural outcomes at 54 months. They found that children attending centre-based ECE at 24 months had significantly lower odds of mother-rated peer problems at 54 months compared to those not attending ECE. Similarly, [56] ([56]) analysed the ECLS-B Cohort Data to investigate the factors influencing 2YOs’ SEL and development, emphasising the importance of SEL for school readiness and advocating for the consideration of other factors while scaling 2YOs’ SEL with standardised psychometric tools, such as their ECE experiences. [62] ([62]) used A Quantitative Report from the FinnBrain Birth Cohort Study to examine how different ECE contexts influence 2YOs’ SEL, including childcare quality and ECE contextual factors. This study suggested the further exploration of care types for later childhood development.

Except for the three longitudinal studies that analysed secondary data, the other six quantitative studies all utilised observational assessments through teacher/parent ratings. Among these, five used naturalistic observation and assessments, while only one used video-based observations, i.e., [65] ([65]). In this study, the researchers conducted video-based observations to examine toddlers’ social experiences and learning-related behaviours in ECE settings, by coding and analysing the recorded footage filmed in a natural environment. Notably, two qualitative studies used both video-based and naturalistic observations, namely, [46] ([46]) and [20] ([20]). [46] ([46]) focused on peer rituals at mealtime, using video recordings to capture and analyse the discourse between toddlers, while [20] ([20]) employed both video and naturalistic observation to study toddlers’ co-authored play narratives and their emerging identities of care. Regarding the two other studies with mixed methods, [64] ([64]) used a video method, and [19] ([19]) used a naturalistic observation to assess 2YOs’ social interaction with peers and explore their SEL.

Besides the methods adopted in the studies, theoretical frameworks also played a crucial role in guiding the research on toddlers’ SEL and peer interactions. A range of theoretical perspectives was employed across the studies. As summarised in the chart, the theoretical frameworks of these 13 studies predominantly draw from developmental psychology, socio-cultural theory, and a holistic development perspective. Developmental psychology is the primary focus in studies that emphasise individual growth and cognitive or emotional development, such as social competence ([38]), emotion regulation ([49]), and language skills ([31]). Among them, the nine quantitative studies and one mixed-methods study, i.e., [19] ([19]) were grounded in developmental psychology, as they assess 2-year-olds’ social–emotional behaviours within peer interactions using related scales or behavioural checklists. These assessment tools were developed based on developmental psychology theory to examine both interpersonal behaviours and intrapersonal development. For example, underpinned by the perspective of developmental psychology, the study of [37] ([37]) and [49] ([49]) adopted quasi-experimental approaches to investigate teacher-led interventions aimed at enhancing 2YOs’ emotional learning and social cognition. These studies both used the CBCL standardised assessment scales and analysed social interactions by comparing pre- and post-test results, assessing the impact of the interventions on the toddlers’ social–emotional development.

Furthermore, in alignment with the findings from RQ 1, all seven studies examining 2YOs’ intrapersonal relationships drew on frameworks from developmental psychology and social cognition. Four of these studies, i.e., [37] ([37]), [49] ([49]), [19] ([19]), and [56] ([56]), which focused primarily on individual development, explored how various developmental factors, such as emotional knowledge, language abilities, and social cognition, influenced 2YOs’ SEL and peer interactions. These studies highlighted the significance of individual cognitive and emotional capacities in shaping early social competence and the way children navigate peer relationships. The remaining three studies, i.e., [38] ([38]), [27] ([27]), and [62] ([62]), adopted a more integrated approach, investigating both intrapersonal and interpersonal development. Regarding the examination of the intrapersonal part, these studies utilised developmental psychology perspectives to examine how individual developmental factors, including emotional regulation and social understanding, influenced SEL within the context of peer interactions.

In addition, both of the qualitative studies adopted a socio-cultural lens to explore 2YOs’ SEL within peer interactions, which focused on the co-construction of the social environment between 2YOs and their peers. [46] ([46]) observed and interpreted 2YOs’ social interactions by video recording and conducted a teacher discussion. In this study, 2YOs’ togetherness and otherness patterns of daily routine were viewed as a phenomenon emerging repeatedly in ECE settings, shaped by children’s lived experiences and knowledge and values. Furthermore, through the socio-cultural perspectives, 2YOs were suggested to contribute to their own peer cultures and belongings by social co-construction with their peers. In turn, the co-constructed ritual routines were examined to facilitate 2YOs’ cooperation and learning with peers. Similarly, [20] ([20]) explored how toddlers’ SE development is influenced by their interactions within peer groups, emphasising the role of cultural practices and shared experiences in shaping SEL. Their findings highlighted the importance of collaborative activities in ECEC settings, where toddlers jointly create and negotiate their social worlds, further underscoring the socio-cultural processes that underpin SEL.

Moreover, one of the two studies that used a mixed-methods approach also took a socio-cultural perspective. [64] ([64]) integrated cognitive development views with constructivism to examine and emphasise the association between 2YOs’ SE skills and cognitive development and the social practices within groups of peers. As shown, while underpinned by developmental perspectives, this study has been complemented by additional theoretical frameworks to explore the complexity of 2YOs’ SEL. Through the socio-cultural perspective, these three studies investigated the nuanced and dynamic factors that influence 2YOs’ SEL, and also highlighted 2YOs’ active role in meaning-making during social interactions with peers.

In conclusion, the majority of studies employed quantitative methodologies grounded in developmental psychology, utilising standardised instruments to assess 2YOs’ SEL, with an emphasis on emotional regulation, social competence, and social behaviours. These studies provided empirical data on developmental trajectories through structured, measurable assessments. Two qualitative studies used a video-based analysis to offer in-depth interpretations of the co-constructive processes between 2YOs and their peers. Additionally, two mixed-methods studies integrated both quantitative and qualitative data within a holistic theoretical framework. Building on these findings, the next section will delve into the role of peers in 2YOs’ SEL.

RQ 3: How does the current research inform our understanding of the role of peers in 2YOs’ SEL in the ECEC context?

This section aims to address RQ 3, which explores the role of peers in 2YOs’ SEL. Building on the findings discussed in RQ 1, which examined the conceptualisation of 2YOs’ SEL in relation to peer interaction, and RQ 2, which focused on the methodologies and theoretical frameworks of the studies, this section synthesises the various roles that peers play in 2YOs’ SEL. This discussion unfolded the role of peers into several key functions, such as facilitators, co-constructors, and predictors or contextual factors, based on the different methodological approaches and findings of the studies (see Table 5).

As defined by frameworks such as the American Academy of Pediatrics ([1]) and the World Health Organisation ([73]), the term toddler generally refers to children between the ages of 1 and 3 years. This age group overlaps with terms like infant and young children, but, typically, toddlers are considered to be in this specific age range. All of the 13 studies reviewed here explore same-age peer interactions in toddler classrooms, and consistently focus on how 2-year-olds interact with peers of the same age, reflecting the significance of same-aged peer interactions for promoting social competence, emotional regulation, and prosocial behaviours.

In four quantitative studies, including one mixed-methods study, i.e., [19] ([19]), peers play a critical role in facilitating 2YOs’ SEL and developmental trajectories. Studies like [42] ([42]), [31] ([31]), [38] ([38]), and [19] ([19]) assessed how peer interactions in ECE settings influence toddlers’ emotional regulation, social competence, and prosocial behaviours. These studies, which rely on observational assessments and teacher ratings, highlighted how toddlers interact with peers and develop SE competency, thereby fostering essential skills such as cooperation, sharing, and emotional regulation. The naturalistic observations in settings like classrooms demonstrated how peers model appropriate social conduct, helping toddlers internalise these behaviours. In these studies, peer interactions are integral to SEL and development, with peers acting as powerful facilitators of both emotional and social competencies. Moreover, the other two qualitative studies, including [37] ([37]) and [49] ([49]), also emphasising the role of peers in facilitating 2YOs’ SEL, while teachers’ intervention on either the interactive environment or 2YOs’ SE knowledge was the focus of these studies.

Four studies, [46] ([46]), [20] ([20]), [64] ([64]), and [65] ([65]) used video-based observations to explore peer interactions in greater depth, positioning peers as co-constructors of 2-year-olds’ development. In these studies, peer interactions are not just observed but actively analysed for how toddlers collaborate in meaning-making, identity formation, and social norms building. For example, [46] ([46]) focused on mealtime rituals, where peers act as co-constructors of shared social behaviours, reinforcing group identity and social belonging through repeated interactions. Similarly, [20] ([20]) examined how toddlers, through peer play narratives, construct identities of care. Video recordings allow for a closer look at how peer interactions foster these narratives, where toddlers negotiate roles and build their positions in collaboration with others. These studies underscore the idea that peer interactions are not just reactive but interactive and collaborative, as peers actively shape each other’s learning, emotions, and social behaviours.

Three studies, [62] ([62]), [56] ([56]), and [27] ([27]), used secondary data analysis to explore how peer interactions influence toddlers’ social-emotional development. In all three studies, peers are seen as both predictors of toddlers’ behaviours and as contextual factors in SEL. [56] ([56]), using ECLS-B cohort data, found that peer interactions in early childhood settings shaped outcomes like self-regulation and school readiness. Similarly, [27] ([27]) showed that peer interactions in preschool settings were linked to behaviours like aggression, cooperation, and empathy. [62] ([62]), focusing on Finnish early childhood education, found that collaborative play and conflict resolution with peers enhanced emotional regulation and social competence. These studies collectively highlight the crucial role of peer interactions in toddlers’ long-term developmental outcomes, particularly in shaping their social–emotional skills, including emotion regulation and prosocial behaviour.

Across these 13 studies, the role of peers in the 2YOs’ SEL is multifaceted, ranging from role models and facilitators to co-constructors and contextual factors. In quantitative studies, peers are primarily viewed as facilitators who provide a reference for important social skills, such as emotion regulation and social competence. In contrast, video-based studies offer more nuanced analyses, positioning peers as co-constructors of social identities and learning. Secondary data analysis studies, on the other hand, highlight how peer interactions predict toddlers’ behaviours and act as contextual factors in their social–emotional development. Together, these studies emphasise the crucial role of peer relationships in shaping toddlers’ developmental pathways, suggesting that ECE programmes should prioritise fostering positive peer interactions to support both inter- and intrapersonal growth. In the next section, these findings will be further integrated, critically analysed, and discussed in relation to their implications for 2YOs’ SEL in ECE contexts.

## 4. Discussion

In this scoping review, we identified 13 relevant empirical studies published between 2014 and 2025 that examined the 2YOs’ SEL in relation to peer interactions in the context of ECE. These studies examined SEL as a broad concept, without delving into specific SEL subdomains. Our findings highlight a gap in research on peer interactions within 2YOs’ SEL, despite the growing recognition of the crucial role peers play and the importance of ECE experiences in their development. Additionally, we found that both the methodology and theoretical perspectives significantly shaped how SEL was conceptualised and how peer interactions were examined. Notably, only two studies adopted a purely qualitative approach to investigate and interpret 2YOs’ peer interactions. Furthermore, most quantitative studies were grounded in developmental psychology and social cognition, focusing on how 2YOs’ intrapersonal social–emotional skills influence their peer interactions. However, several studies have emphasised the need to consider additional social factors, such as the type of peer setting and individual ECE experiences, in shaping 2YOs’ SEL.

In the process of screening papers and examining selected papers, it is also identified that there is a tendency to foreground adult-facilitated or teacher-led strategies for fostering 2YOs’ SEL and peer interactions, often underestimating peer-focused dynamics and the spontaneous nature of toddler peer interactions ([21]; [24]; [67]). [40] ([40]) have argued that teacher influence on 2YOs’ SE development is relatively limited, especially when broader contextual factors and peer relationships are taken into account. Similarly, research by [28] ([28]) highlighted the richness of peer interactions as fertile grounds for developing emotional understanding, cooperation, and self-regulation, often outside of adult scaffolding. This suggests a need to shift attention toward the peer context as a primary factor for early SEL development.

Regarding the conceptualisation of 2YOs’ SEL, they varied depending on the theoretical lens adopted. Studies grounded in a developmental perspective typically conceptualised SEL in terms of 2YOs’ cognitive and emotional competencies, which were often associated with school readiness and stage-based milestones, such as [56] ([56]). These studies usually employed well-established instruments to assess children’s social and emotional behaviours with a quantitative approach, such as [19] ([19]). While these approaches provided measurable indicators for measuring 2YOs’ SE behaviours within peer interactions, they may overlook the richness of their social experiences. In contrast, combined with other perspectives, including sociocultural theory and constructivism, 2YOs’ SEL within peer interactions can be conceptualised as a multi-dimensional and dynamic relational process shaped by both 2YOs and their peers. These perspectives emphasised the co-construction of meanings and the reciprocal nature of 2YOs’ peer interactions. For instance, [46] ([46]) and [20] ([20]) explored how 2YOs actively utilised the environment and inter-personal sources to engage in mutual meaning-making.

Furthermore, by taking a qualitative approach and video methods, studies such as [46] ([46]) and [20] ([20]) focused on interpreting the situated meanings of children’s actions within peer interactions, rather than relying on predetermined checklists or developmental frameworks to assess 2YOs’ social behaviours in a quantitative way. Predetermined assessments often constrain exploration within a monologic and static framework, overlooking the complexity and individuality of children’s experiences. They assume universal developmental norms and risk reducing children’s behaviour to fixed categories, rather than appreciating how personality and lived experience shape each child’s unique ways of engaging socially. Drawing on the perspective of [3] ([3]), meaning does not pre-exist in behaviour but emerges through the interaction of voices between children. This interpretive stance may support the study to generate nuanced, context-sensitive insights into how young children navigate and express their social orientations in peer settings.

Another important finding is that 2YOs’ position within peer interaction varies based on theoretical perspectives. In studies assessing 2YOs’ SEL within peer interactions viewed through the lens of developmental psychology, peers were often taken as 2YOs’ development facilitators. In turn, 2YOs often take passive positions as inferior socialisers, rather than being viewed as co-constructors of social events. For example, older peers were commonly viewed as scaffolding agents who supported younger children’s SEL through guided participation ([64]), drawing on [74]’s ([74]) Zone of Proximal Development. This perspective tends to emphasise a unidirectional model of learning in which 2YOs require guidance from more competent others to participate meaningfully in peer interactions.

However, such perspectives have been critiqued for underestimating the nuanced social dialogues and SEL competencies that even very young children demonstrate in social interactions. The relational approaches emphasise the co-construction of meaning, showing that 2YOs are capable of initiating, sustaining, and shaping peer interactions in ways that reveal critical social agency ([6]; [29]; [35]; [70]). Peer agency, often underrepresented in traditional developmental models, plays a significant role here, as 2YOs actively influence and contribute to the direction of their social interactions, rather than merely reacting to adult or peer guidance. In practice, though, peer agency is frequently overshadowed by pedagogical factors, such as teacher–child ratios and national curricula, rather than being shaped by the developmental characteristics of 2YOs and their active role in peer interactions ([33]). Instead of taking the view that learning is a unidirectional process in which 2YOs internalise knowledge through language exchanges ([66]), 2YOs’ SEL within peer interactions can be seen as a dynamic, bidirectional process within dialogic environments.

Overall, the findings suggested that the existing research should benefit from embracing more diverse perspectives that recognise the active role of 2YOs in shaping their SEL experiences. 2YOs’ social dialogues have been suggested to be fluid, complex, and multi-level ([15]; [55]). Instead of depending on a singular perspective or pre-determined measures, future research should explore how 2YOs engage in peer interactions and their nuanced expressions of SEL. This will support a more comprehensive and in-depth understanding of SEL as both an individual and collective developmental process.

## 5. Implications

Diverse theoretical perspectives are suggested to be adopted in the future to explore 2YOs’ SEL within peer interactions. The findings of this scoping review provide insights into how theoretical perspectives impact the way of understanding and investigating 2YOs’ SEL and the role of peers. The review reveals that the postivism approach, grounded in developmental psychology perspectives, has been predominantly adopted in existing research. As a result, prevailing approaches to investigating general SEL may be misaligned with more nuanced ways of understanding 2YOs’ lived experiences of SEL, particularly in the context of peer interactions. Since children’s social interactions are often fleeting, it is challenging to capture their complex unfolding, fluidity, and multi-level nature ([9]; [15]); multi-faceted perspectives should be taken to capture their complex social interactions, such as the use of visual methods ([17]), developing age-responsive pedagogy ([69]; [71]), investigating young learners’ groupingness ([10]), and the examination of toddlers’ peer interaction patterns ([43]).

In addition, multi-dimensional study approaches could be used to capture 2YOs’ nuanced social and emotional manifestations. As a whole-body speaker ([43]), 2YOs’ social experiences within SEL call for multi-dimensional approaches to their meaning-making and learning process ([13]). Visual modes of inquiry could be particularly beneficial in this context since they provide ways of analysing language in movement ([54]). These methods are suggested to capture fleeting moments, unfold the complexity of social dialogues, and provide opportunities to generate new thoughts from existing knowledge ([55]; [69]). Moreover, in the context of ECE, a “post-developmental methodology” has been suggested, which is beyond the perspective of ages and developmental stages for the research targeting very young children ([36]). Accordingly, future research should adopt diverse perspectives and methodological approaches that provide more comprehensive insights into 2YOs’ SEL and their peer interactions.

## 6. Conclusions

This scoping review synthesised the existing literature published from 2014 to 2025 regarding the conceptualisation of 2YOs’ SEL, the methodology and theoretical perspectives, and the role of peers in their SEL in the ECE context. This literature review reveals the dominant methodological approaches that unfold 2YOs’ SEL within peer interactions, which highlights a gap in using diverse methodologies on this topic. Most existing studies predominantly employ positivist methodologies and adopt predetermined checklists based on the perspective of developmental psychology. This approach has contributed to an emphasis on teacher-led SEL initiatives and systemic assessment of 2YOs’ SEL; however, it limits the exploration of peer-driven SEL and 2YOs’ agency in peer interactions. In this approach, peer interactions are always measured according to a standardised behavioural checklist in a static and monological way, with limited consideration of dynamic and context-based social encounters.

Although 2YOs are situated in peer-rich environments in the context of ECE and some quantitative studies have recognised them as active contributors to the social construction of SEL, 2YOs’ active roles in the construction of the social environment and dynamic relationships with peers have been suggested to be underestimated according to the scoping review. This reflects a tendency to portray 2YOs as passive recipients and treat peers as the behavioural reflectors of 2YOs, though some studies take peers as one of the critical environmental factors. The scoping review indicates that the field of SEL research for this age group appears to lag behind the broader trends in ECE scholarship, which emphasise the reciprocal and situated meaning-making in social interactions. Therefore, a shift toward recognising the mutual, embodied, and socially constructed nature of peer interactions may provide richer and more comprehensive insights into the SEL of 2YOs ([68]).

## 7. Limitations

The scoping review has some limitations. Firstly, it primarily focused on studies that examined the overall SEL in 2YOs and did not fully explore the broader scope of 2-year-old peer social competence within ECE contexts. While many other studies have investigated specific aspects of SEL, such as relationship building or emotional expressions in peer interactions, they were excluded from the review because they did not explicitly use the term “SEL” (e.g., [7]; [25]). Including these studies could provide valuable insights into the diverse perspectives and methodologies researchers have used to understand SEL in this age group and context.

Accordingly, the exclusion of studies not explicitly using the term “SEL” may have introduced a methodological bias, limiting the ability to capture the full range of research approaches. This narrow focus could also result in an overemphasis on studies that define SEL in a particular way, potentially overlooking more nuanced understandings of social and emotional development. Furthermore, this selection bias might have affected the inclusion of studies with alternative theoretical frameworks, which could provide complementary or contrasting views to the mainstream SEL literature. Therefore, including a wider array of studies, regardless of the terminology used, could enrich the understanding of how peer interactions shape 2-year-olds’ social competence in ECE settings. Additionally, the selection of a specific year range (2014–2025) for the literature may have further introduced bias, potentially overlooking earlier foundational studies that laid the groundwork for more recent developments in this field. Therefore, including a wider array of studies, regardless of the terminology or the year range used, could help uncover more related work and enrich the overall understanding of the field.

Furthermore, future reviews should consider incorporating databases that are published in languages other than English and representing diverse cultural contexts to enhance the comprehensiveness of the evidence base. Although the current scoping review did not adopt any language restrictions during the search, the results revealed a pronounced predominance of English and Western cultural contexts. This indicates a potential language and culture bias that may limit the representation of research conducted in other linguistic and cultural contexts.

## Figures and Tables

**Figure 1 behavsci-15-01303-f001:**
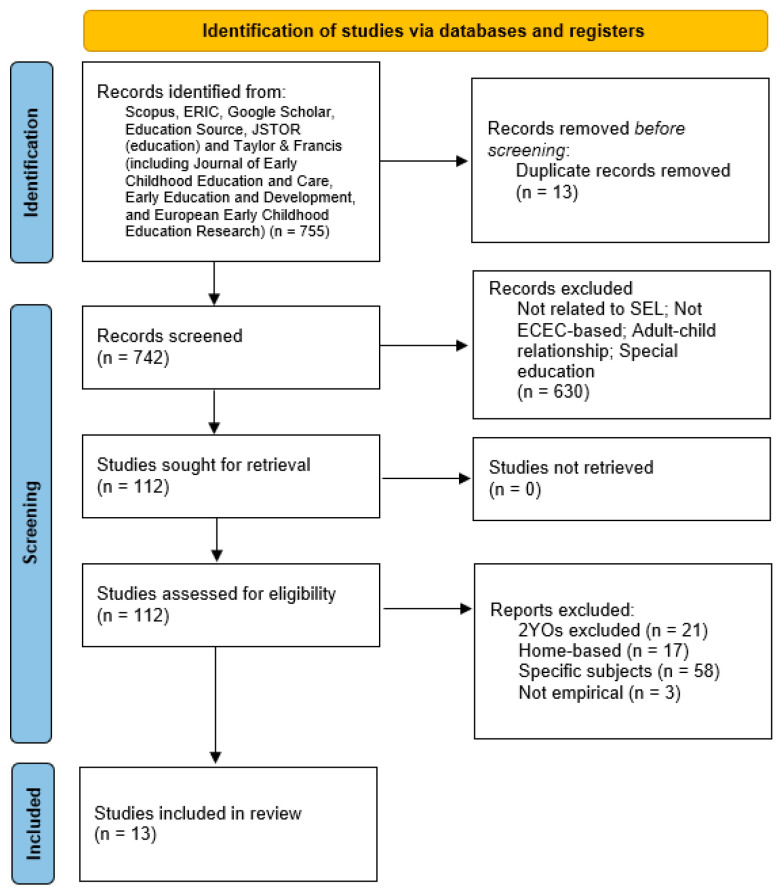
Flow chart of the study selection.

**Table 1 behavsci-15-01303-t001:** Inclusion and exclusion criteria.

Criteria	Inclusion
Participants	Studies that specifically incorporate two-year-olds.
Source types	Peer-reviewed journal articles with full-text availability; Empirical studies.
Year of publication	Studies are published between 2014 and 2025.
Field of knowledge	Studies are situated within the context of ECE; Studies are situated within the subject of Education; Studies take SEL as the unified concept, rather than specific subjects; Studies explored 2YOs’ peer interaction, not child–teacher relationships.

**Table 2 behavsci-15-01303-t002:** Summary of study characteristics.

Study	Children Age	Sample Size	Country	Empirical Type
[42] ([42])	11–41 months	93 classrooms	USA	Primary data
[37] ([37])	2-year-olds	57	Spain	Primary data
[46] ([46])	8–20 months	4	NZ	Primary data
[31] ([31])	33 months	1005	Norway	Primary data
[49] ([49])	26–36 months	95	Italy	Primary data
[19] ([19])	24–47 months	149	Italy	Primary data
[38] ([38])	30–33 months	1084	Norway	Primary data
[64] ([64])	28–45 months	120	Netherlands	Primary data
[20] ([20])	15–23 months	9	NZ&AUS	Primary data
[27] ([27])	Toddlers to preschool	6536	NZ	Secondary data
[65] ([65])	12–36 months	130	Germany	Primary data
[56] ([56])	9 months–5.5 years	4950	USA	Secondary data
[62] ([62])	2YOs	1104	Finland	Secondary data

**Table 3 behavsci-15-01303-t003:** SEL in selected studies.

Focus on SEL	Study	Examination in SEL	Teachers’ Role
Inter-personal	[42] ([42])	Relation between classroom quality and 2YOs’ social behaviour problems	Quality of the environment built by teachers
	[46] ([46])	Co-construction with peers for daily routines as rituals	Children’s co-construction of meanings
	[31] ([31])	Peer interaction is more influenced by interpersonal skills than language skills.	Free play by children
	[64] ([64])	Group-centred support on 2YOs’ collaborative play and SEL	Group-based free play
	[20] ([20])	Caring acts and empathy in social play	Children’s co-construction of meanings in play and caring
	[65] ([65])	Learning-related behaviours in peer interactions	Learning-related behaviours in peer interactions
Inter and intra	[38] ([38])	Holistic development of well-being, play, and social competence in peer settings	Child-led free play
	[27] ([27])	Multiple inter- and intra-personal skills are assessed to examine the impact of ECE attendance duration	Quality of the environment built by teachers
	[62] ([62])	SEL is shaped by multiple environmental and developmental influences in childcare settings	Quality of the environment built by teachers
Intra-personal	[37] ([37])	Influence of social cognition/emotional knowledge/self-regulation/theory of mind on social outcomes	Interventions from teachers
	[49] ([49])		Training led by teachers
	[19] ([19])		Quality of the environment built by teachers
	[56] ([56])		Quality of the environment built by teachers

**Table 4 behavsci-15-01303-t004:** Methodologies and theoretical frameworks in selected data.

Study	Paradigm	Method	Tool Used & Country	Theoretical Framework
[42] ([42])	Quantitative	Structured observation Teacher ratings	CLASS-Toddler and ITERS-R	Developmental psychology
[37] ([37])	Quantitative Intervention study	Pre-/post-test, observation, teacher ratings	Child Behavioural Checklist (CBCL)	Developmental psychology Emotional theory
[46] ([46])	Qualitative Ethnographic	Video observation Discourse analysis	N/A	Sociocultural theory
[31] ([31])	Quantitative correlational study	Observational assessments Teacher ratings	Tras and Alle med (Norway)	Language development theory Social competency theory Developmental psychology
[49] ([49])	Quantitative Quasi-experimental	Pre-/post-test, observation, teacher ratings	Social Competence and Behavioural Scales (adapted from CBCL)	Developmental psychology Theory of mind Social cognition
[19] ([19])	Mixed method	Structured observation Teacher ratings Teachers’ interview	Affect Knowledge Test Diverse-Desire Task/True-Belief Task Peabody Picture Vocabulary Test—Revised	Developmental psychology Social cognition
[38] ([38])	Quantitative	Observation Teaching rating	Alle med (Norway)	Whole-child perspective Play theory
[64] ([64])	Mixed method	Group-centred video observation and assessment: thematic analysis Code and quantify group interactions	Classroom Assessment Scoring System (CLASS) Toddler	Sociocultural theory Constructivism
[20] ([20])	Qualitative	Video observations Narrative analysis	N/A	Sociocultural theory Care Ethics
[27] ([27])	Quantitative Longitudinal Secondary data	Longitudinal cohort data, assessment tool, and teacher ratings	Strengths and Difficulties Questionnaire Growing Up in New Zealand longitudinal birth cohort	Developmental psychology Socio-cultural theory
[65] ([65])	Quantitative	Video observation Teacher ratings Statistical analysis for correlations	Activation of learning potential (ALP) Attachment-Q-sort	Developmental psychology Sociocultural theory
[56] ([56])	Quantitative Longitudinal Secondary data	Teacher and parent ratings Statistical modelling	National cohort data and SE items (ECLS-B)	Developmental psychology
[62] ([62])	Quantitative Longitudinal Secondary data	Teacher and parent ratings Survey and analysis of secondary data	Brief Infant–Toddler Social and Emotional Assessment (BITSEA) FinnBrain Birth Cohort Study	Bioecological model Developmental psychology

**Table 5 behavsci-15-01303-t005:** Characteristics of peers in selected studies.

Study	Peer Age	How Peer Interactions Were Assessed	SEL Variables
[42] ([42])	Same age	Observational tools (CLASS-T, ITERS-R)	Cooperative behaviours; emotional expression
[37] ([37])	Same age	Teacher-led interventions and observations of children’s peer interactions during structured activities	Emotional knowledge; empathy; prosocial behaviours
[46] ([46])	Same age	Video-based analysis of mealtime rituals and peer interactions	Peer rituals, social cooperation
[31] ([31])	Same age	Observational assessment of play behaviour and peer interactions in natural settings	Play behaviour; communication skills
[49] ([49])	Same age	Pre- and post-test observational assessment of prosocial and aggressive behaviours during peer interactions	Emotional knowledge; prosocial and aggressive behaviours
[19] ([19])	Same age	Teacher interviews and observations of children’s social interactions during free play and structured activities	Social cognition; prosocial behaviour; language development
[38] ([38])	Same age	Observational assessment of free play and peer interactions in natural settings	Social competence; emotional expression; movement skills
[64] ([64])	Same age	Observational analysis of peer interactions within group activities in the ECE setting	Group dynamics; peer cooperation; conflict resolution; social bonding
[20] ([20])	Same age	Video-based observation of play narratives and peer interactions	Caregiving behaviour; peer play narratives; social roles
[27] ([27])	Same age	Peer interactions were assessed through structured observations from the secondary dataset and behavioural ratings from teachers	Social behaviour outcomes; emotion outcomes
[65] ([65])	Same age	Video-based observations of group-based peer interactions during play and learning activities	Learning-related behaviour; peer engagement percentage of time
[56] ([56])	Same age	Secondary analysis of ECLS-B cohort data, which includes parent and teacher ratings of peer interactions and behaviours	Emotion regulation; interaction social behaviour
[62] ([62])	Same age	Teacher and parent survey, and the secondary data analysis	Peer interaction quality; SE behaviours

## Data Availability

No new data were created.

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
