# Peer review of "Social–Emotional Learning of 2-Year-Olds Within Peer Interactions in Early Childhood Education Settings: A Scoping Review"

_behavsci, 2025, doi:10.3390/bs15101303_

Round 1
Reviewer 1 Report
Comments and Suggestions for Authors
REVIEW REPORT
Title: The title is clear regarding the main topic: a systematic review of the literature on the absence of socio-emotional learning in 2-year-old children in ECEC. This helps readers immediately understand the focus of the article. However, it is somewhat lengthy. It should be simplified and made more manageable and attractive without losing specificity. The abbreviation ECEC should be explained to make it more comprehensible for the academically interested audience.
Abstract: The abstract introduces the topic and its necessity but lacks clarity in the formulation of objectives (they are very vague) and the methodology used, particularly regarding the inclusion and exclusion criteria necessary to assess the suitability of the sources providing the data to meet the research objectives. It is also inappropriate to include unexplained abbreviations, as this reduces the clarity of the content. There is a lack of precision in the number of sources and the time period covered.
Key words: Ensure that they are indexed in educational thesauri.
Introduction: The topic and its necessity are well justified.
Research Questions: The research questions are presented in a disordered manner, in relation to what is indicated in the abstract, and the logical order that, in my opinion, should be followed: first, the definition should be deepened, second, the methodological approaches, and finally, the research on the topic. The first provides a clear and specific conceptual basis for the study, ensuring that all readers understand what SEL is in this context. The second allows understanding the approaches used to contextualise the studies and their findings. After defining SEL and understanding the methods, exploring how peers influence the socio-emotional development of 2-year-old children is crucial to understanding interpersonal dynamics and their impact.
Methodology: There are significant issues that are difficult to address without substantial changes. Although relevant databases have been used, to ensure broader coverage of the literature, it would have been important to expand them with others such as WOS, or more specialised in education and child development, such as PsycINFO or JSTOR. The authors should have adjusted the inclusion and exclusion criteria to ensure they are sufficiently specific and relevant, guaranteeing that only the most pertinent and high-quality studies were included in the review. For the evaluation of the quality of the studies, tools such as the STROBE checklist for observational studies or the CONSORT checklist for randomised controlled trials could have been used to assess the validity and reliability of the studies. It would also have been necessary to conduct a bias analysis (publication bias, selection bias, etc.) in the selected studies. A positive point has been the use of tables and figures to summarise the findings visually and accessibly, helping readers better understand the results and identified trends. Additionally, all steps of the review process should be indicated, from the initial search to the synthesis of results. This includes the search strategy, selection criteria, and quality evaluation process. Transparency in these aspects improves the reproducibility of your review. However, the keywords used to extract the necessary aspects to meet the research objectives have not been documented or justified precisely and sufficiently.
Results: The results are adequate and address the research objectives. They have limitations derived from the previously mentioned methodological issues.
Discussion: The fact that the review includes only studies published in English limits the analysis of the topic from different cultural contexts. Limiting to English may introduce language bias, excluding relevant and valuable research published in other languages. Not limiting the search to English ensures that the discussion is broader and richer, considering how these contexts may influence the results and applicability of the findings. Including studies in other languages can enrich the review with diverse perspectives and methodological approaches that may not be present in the English literature. Implementing these improvements can strengthen the quality and relevance of your systematic review, providing a solid foundation for future research and practical applications.
Conclusions, Limitations, and Prospective: There is a lack of precise conclusions regarding the scope of the results and a critical analysis of the limitations and future directions. It would be beneficial to include a deeper discussion on the study's limitations, potential biases, and suggestions for future research based on the literature review.
References: The references are adequate and recent, and can be enriched with others that broaden the view of the topic from other cultural and linguistic perspectives different from English.
Decision: The work has methodological shortcomings that require significant modifications to increase the interest, quality, and scope of the results. These relate to the small and unrepresentative number of sources used, the search strategy (it is necessary to expand the databases used, specify the search terms more accurately, justify the applied filters, etc.), the evaluation of the quality of the included studies, bias analysis, and how the results synthesis was conducted (qualitative and quantitative analysis). For these reasons, my decision is REJECTION.
Comments on the Quality of English LanguageThe quality of the English language in the text could be improved. I have noticed some stylistic errors that affect the clarity of the article. I recommend considering a more thorough review or the implementation of an additional proofreading process to ensure that the texts meet the expected quality standards.
Author Response
Please kindly check the attachment.

Reviewer 2 Report
Comments and Suggestions for Authors
Thank you the opportunity to review this paper that provides an overview of current research into 2-year-olds social-emotional learning within peer interactions in ECEC. This is certainly an important area of investigation and I commend the research team for considering the research on this topic at this early, important and often overlooked age-level in ECEC research.
Title
- The title suggests that there is an absence of social-emotional learning at this age level. My understanding is that the authors are not suggesting this but rather that there is an absence of research investigating SEL at this age level within peer interactions in ECEC. Perhaps the words 'the absence' could be removed from the title.
General comments that might need some signposting/clarity throughout the paper
- So by including empirical research in the title does this mean that all papers included in the review included an SEL outcome variable?
- The paper seems to read more as a scoping than a systematic review to me. There is no measure of bias (use of a quality assessment tool) and the goals seem to be to scope the current evidence on a topic and identify conceptual and research gaps rather than to synthesise research answer a specific question per se.
Munn, Z., Peters, M. D. J., Stern, C., Tufanaru, C., McArthur, A., & Aromataris, E. (2018). Systematic review or scoping review? Guidance for authors when choosing between a systematic or scoping review approach. BMC Medical Research Methodology, 18(1), 143–143. https://doi.org/10.1186/s12874-018-0611-x - Did all of the papers include 'peer interactions' in their analyses as a predictor variable or a context in which SEL was examined, or a key component of an SEL intervention that was examined, or all of these?
- Are peer interactions here all same aged child interactions, group interactions, interactions with older peers, or all of these?
- Given that the paper is reviewing some empirical papers I would be included to avoid the word 'significance' when referring to importance so that it is not interpreted as a significant effect.
Introduction
- Following the first line, it would be useful to provide a brief description of SEL as conceptualised by the researchers. Although the focus of the article is to provide an overview of definitions of SEL in the included research it would be useful here to provide a broad definition that specifies the researchers point of view given that it is a big concept.
- page1 lines 40-43 are reading as though 2 years is that 2 years is too early for SEL to take place. Rather there is a large literature indicating that SEL is taking place in iteractions from birth. It think this is not the authors point of view and rather what they are keen to say is that is that some researchers have suggested that investigating SEL within peer interaction in ECEC at age 2 has been limited by the perspective that 2 year olds may not benefit from interaction with their peers to the same extent as older children.
- Great examples across lines 44 to 50 as to why this age level may have been overlooked.
- page 2 lines 51 to 66 provide a good overview of SEL. It would be good to include more on specific skills considered to be SEL or a visual of such.
- page 2 line 69 specify 'an emphasis on 2-year-olds cognitive capabilities may lead researcher to ..'
- page 2 line 72 what is meant by 'great association' - does this mean a strong positive effect/ strong significant (statistical)
- In regard to the lack of research in this area for 2 year olds it might also relate to (1) less research on this age level in ECEC generally. This might be worth noting in the intro or discussion.
- Line 89 what is meant by broader contexts?
- line 89 - By 'highlight of cognition' do you mean the 'perception of lower cognitive capability for SEL in the context of peers'
- My preference would be to present the definitions before the methodlogical approaches - as qs 1 and 3 flow into each other well.
- In the intro it could be beneficial to make a clear statement on the importance of considering SEL in peer interactions at this age. What potentially could peer interactions afford SEL at this age. Research with siblings could be useful to reflect upon here.
Methods
- See point on scoping review earlier and general points about what is/is not considered in this review
- Were searches conducted in database on full-text, title and abstract or key words.
- There are very few search terms. Were any other related terms for concepts used? (again this might be a good reason to call this a scoping review)
- In regards to terms SEL is a large overarching concept that includes cognitive and social concepts (as is evident in the definitions). I understand given the aim of the review to keep the search specific to this term. It would be useful to make a statement on the reason for searching for papers that include this broad definition only rather than the many components of SEL.
- I have not seen a parallel search approach previously. Why only include ERIC on the round 2 search?
- page 4 lines 172-173. This is a little confusing. Is it the case that only studies that specified their topic areas as spanning multiple disciplines were included and that studies that specified one disciple were excluded. Alternatively is it the case that research was considered from all of the listed relevant disciplines?
- Page 7 line 268 - check for typo/wording
- page 8 lines 307-309. The aim of the paper is very clear here. I would also use this wording earlier in the intro to more clearly specify the aim of the study.
Results
Throughout the results there were a few things that remained unclear to me that might require clarity or signposting.
- In each study were the peers same-age or older (i.e., how were peers defined)
- Was SEL measured in each study and how?
- In each study was SEL in peer interactions explicitly measured and evaluated, a component of an intervention or merely commented on. This is not clear and makes it difficult to interpret the relevance of the findings.
- Why include the meta-analysis since it has likely reviewed similar studies?
- I would report on the definitions before methodological approaches and findings. The definitions/perspectives have been broadly grouped. For each of these groupings it would be useful for the authors to describe what is meant (from their perspective) by each domain/perspective term (e.g., how is socio-cultural perspective defined?). For example the authors have listed the studies that take a developmental lens in their approach to 2yo SEL and provide some examples of this perspective but do not explain what this perspective is at the opening of the paragraph.
- I wonder if research qs 2 and 3 could be combined in the results section? Perhaps the papers with similar methodological approaches could be discussed under subheadings and the key findings and implications outlined in this format. I think it would be easier to follow as I found myself moving back and forth between these sections to piece this information together.
Discussion
I disagree that a developmental approach sees/assumes children as being incompetent SE learners. Children are developing increasingly complex SE skills from birth and this is acknowledged in developmental theory and research. Rather those studies that have focused primarily on developmental stages in SEL may fail to consider or examine the role of the child's context in shaping this development to the same extent as those taking a more socio-cultural approach. Indeed peer interactions may support/hinder aspects of social-emotional learning via many mechanisms which are yet to be examined in research (future direction) and certainly there may be specific compositions within a peer group (e.g., older/younger, gender composition, developmental profiles) that support/hinder certain learnings. In some cases the educators' interactions with the group may also support/hinder perspective taking (it is likely multidimensional).
Author Response
Please kindly check the attachment.

Reviewer 3 Report
Comments and Suggestions for Authors
This paper conducts a systematic literature review to examine the current state of research on social-emotional learning (SEL) among two-year-old children (2YOs) within early childhood education and care (ECEC) settings, particularly through peer interactions. The chosen topic holds significant theoretical value and practical relevance. Given the rapidly increasing global enrollment of 2YOs in ECEC programs, focusing on the social-emotional experiences of this specific age group is both highly timely and innovative. The overall structure of the paper is coherent, the literature search and selection process is methodologically sound following the PRISMA framework, and the arguments are logically organized, with rigorous analysis and strong conclusions that contribute meaningfully to the field.
However, several aspects could be further improved:
- Enhancement of the Introduction Section
The current introduction acknowledges the insufficient attention given to 2YOs’ SEL research but provides only a brief explanation. It does not fully and systematically elaborate why existing studies predominantly focus on children older than two, which somewhat weakens the articulation of this review’s significance. To strengthen the rationale and originality of the study, it is recommended to enrich the introduction by:Systematically explaining the reasons behind the neglect of the two-year-old group, including:-
- Underestimation of 2YOs’ capabilities by traditional developmental stage theories (e.g., Piaget, Vygotsky);
- Existing SEL assessment tools are mostly designed for children aged three and above, lacking adaptations for 2YOs;
- The prevalent caregiving-oriented positioning of 2YOs in educational practice, resulting in insufficient research and intervention efforts targeting their social-emotional development. Emphasizing the unique developmental importance of the two-year-old stage, particularly the rapid growth in empathy, emotion regulation, and emerging self-awareness, highlighting that two years is a sensitive period for social development rather than simply an extension of infancy.
- Highlighting the consequences of neglecting this age group, such as missed critical intervention windows that could negatively affect long-term social adjustment and mental health outcomes.
- Naturally leading into the contribution of the current study by positioning it as an important effort to fill this critical gap and deepen the understanding within the field.
-
- Discussion of Search Limitations
The paper restricts its literature search to English-language studies published between 2013 and 2024. While this is a common practice, the potential bias introduced by these constraints is not sufficiently discussed. It is recommended to explicitly acknowledge and reflect on the possible limitations stemming from these restrictions in the methodology section. - Broadening of Literature Sources
The literature reviewed heavily cites studies from Western countries, particularly the United States and the United Kingdom. Given the cultural dependency of ECEC practices, it is advisable to incorporate more studies from non-English-speaking regions in future work to enhance the generalizability and diversity of the review’s findings. - Improvement of Data Extraction and Analysis
Although the paper employs content analysis for data extraction, the current analysis lacks sufficient granularity. It is suggested to add a section evaluating the methodological quality of each included study, including basic assessments of the instruments used, observational methods, sample sizes, and study designs (e.g., cross-sectional vs. longitudinal). This would allow for a more powerful conclusion, clearly identifying not only the scarcity of research but also specific methodological weaknesses within the existing body of work.
Reviewer 4 Report
Comments and Suggestions for Authors
Paper Title: The Absence of 2-Year-Olds' Social-Emotional Learning within Peer Interactions in ECEC: A Systematic Review of Empirical Research Concerning 2YOs’ SEL.
General comments
The paper is insightful and competent, and it contributes knowledge on 2YOs’ SEL in ECEC contexts and fills a gap in research. The topic is relevant for the field of ECEC.
The author(s) need to do some minor works to further enhance the quality of the paper. Some specific areas that need attention are included here, and comments on the pdf version of the paper.
Title/Abstract
- Modify the title to eliminate repetition and capture the essence of the paper.
Introduction
- Delete the many uses of e.g., in the in-text citations in the entire paper. See L35 and others
- Define ECEC and/or context to provide a clear understanding of 2YOs SEL
- Make sentences like these L73-74; L97-98 and others clearer and more meaningful.
Methodology
- Check L124, L130 and L139 and other comment in this section to clarify or modify appropriately
Results
- Overall check L316, L318 and others under the result section as indicated in the pdf comments to modify, correct or make meaningful
RQ1/RQ2/RQ3
- Make “among them” specific L339
- Rewrite to make clear L440-442
- Reference some of the specific studies L495-497
Discussion
- Excellent discussion
Conclusion/Implications
- Add one specific implication for ECEC programs and practice for 2YOs
References
- Update in-text citations and ensure the reference list suits the preferred style of the journal.

Author Response
Please kindly see the accouchement.

Round 2
Reviewer 1 Report
Comments and Suggestions for Authors
REVIEW REPORT 2
Title: Although it has been modified and shortened, it remains redundant and imprecise in its contribution. The authors should consider choosing either the educational stage where it is developed or the age group to avoid repetition, and better specify the topic and the specific aspects they address.
Abstract: It has been improved and most of the suggestions have been addressed, but there is still a lack of precision regarding the number of sources and the time period covered by the literature review.
Key words: The suggestion has been adequately addressed.
Introduction: The topic and its necessity are well justified. The research questions have been appropriately ordered.
Methodology: There are significant issues that are difficult to resolve and detract from the value of the results. As mentioned in my previous review, although relevant databases have been used, given the specificity of the topic, the authors could have considered adding other specialised databases or additional sources to ensure that relevant studies were not missed. To ensure broader coverage of the literature, it would have been important to expand with others such as WOS, or more specialised in education and child development, such as PsycINFO or JSTOR. The authors should have adjusted the inclusion and exclusion criteria to ensure they were sufficiently specific and relevant, guaranteeing that only the most pertinent and high-quality studies were included in the review. For the evaluation of study quality, tools such as the STROBE checklist for observational studies or the CONSORT checklist for randomised controlled trials could have been used to assess the validity and reliability of the studies. It would also have been necessary to conduct a bias analysis (publication bias, selection bias, etc.) on the selected studies. A positive point has been the use of tables and figures to summarise the findings in a visual and accessible manner, helping readers better understand the results and identified trends. It is also important to indicate all steps of the review process, from the initial search to the synthesis of results. This includes the search strategy, selection criteria, and quality assessment process. Transparency in these aspects improves the reproducibility of the review. However, the keywords used in the extraction of the necessary aspects to fulfil the research objectives have not been documented or justified precisely and sufficiently. In general terms, although the review procedure has been improved, there are aspects that the authors have not been able to address, which detract from the rigour and interest of the results, which are based on only 10 sources, in my view insufficient and unrepresentative of all that may have been published on the topic and the variables of interest.
Results: They are adequate and address the research objectives. They have the limitations derived from the aforementioned methodological issues.
Discussion: The fact that the review includes only studies published in English limits the analysis of the topic from different cultural contexts. Limiting to English may introduce language bias, excluding relevant and valuable research published in other languages. Not limiting the search to English ensures that the discussion is broader and richer, considering how these contexts may influence the results and the applicability of the findings. Including studies in other languages can enrich the review with diverse perspectives and methodological approaches that may not be present in the English literature. Implementing these improvements can strengthen the quality and relevance of your systematic review, providing a solid foundation for future research and practical applications.
Conclusions, limitations, and future directions: There is a lack of precise conclusions regarding the scope of the results, and a critical analysis of the limitations and future directions. It would be beneficial to include a deeper discussion on the study's limitations, possible biases, and suggestions for future research based on the literature review.
References: They are adequate and recent, and can be enriched with others that broaden the perspective of the topic from different cultural and linguistic viewpoints other than English.
Decision: Although the work has been improved from the previous version, it still has the same methodological shortcomings that detract from the interest, quality, and scope of the results. These relate to the small and unrepresentative number of sources used, the search strategy (it is necessary to expand the databases used, specify the search terms more accurately, justify the applied filters...), the evaluation of the quality of the included studies, bias analysis, and how the synthesis of results was conducted (qualitative and quantitative analyses).
For these reasons, I maintain my decision to REJECT.
Round 3
Reviewer 1 Report
Comments and Suggestions for Authors
Manuscript Title:
Social-emotional learning in peer interactions among 2-year-olds: a scoping review in the context of Early Childhood Education
Report Summary
I appreciate the effort made in this revised version of the manuscript. Several of the suggestions raised in the previous review have been addressed, particularly regarding structure, clarity of objectives, and presentation of results. However, methodological limitations remain that affect the validity, representativeness, and scope of the review. Below, I outline the improved aspects and those that still require attention, along with specific recommendations.
- Title
Observation: Although the title has been shortened and the educational context specified, it remains redundant by mentioning both age and educational stage.
Recommendation: Consider referring to either age or stage, not both, and clarify the thematic focus. For example: “Peer interactions and social-emotional learning in 2-year-olds in ECE settings”.
- Abstract
Observation: Clarity has improved, with the review period (2014–2025) and number of studies (13) now included.
Recommendation: Add the total number of initially identified studies, the exclusion criteria applied, and a brief mention of methodological limitations to enhance transparency.
- Keywords
Observation: Appropriately revised and relevant.
- Introduction
Observation: Well-structured, with a clear justification of the topic’s relevance and well-formulated research questions.
Recommendation: No changes required in this section.
- Methodology
Observation: The range of databases has been expanded (including JSTOR and Taylor & Francis), which is a positive development. However, several shortcomings persist: Key databases such as WOS and PsycINFO were not included. Standardised tools for assessing study quality (e.g., STROBE, CONSORT) were not used. No bias analysis (e.g., publication or selection bias) was conducted. The synthesis of results does not specify whether qualitative or quantitative methods were applied.
Recommendations: Broaden the search to include more specialised databases. Incorporate a systematic assessment of the quality of included studies. Conduct and report a bias analysis. Clearly state the synthesis approach (e.g., thematic, narrative, quantitative).
- Results
Observation: Well-organised and aligned with the research questions. Detailed and helpful tables are provided.
Limitation: The number of studies (13) remains limited for a scoping review, which affects representativeness.
Recommendation: Provide a stronger justification for the final selection and consider expanding the corpus if inclusion criteria are made more flexible.
- Discussion
Observation: The limitation of including only English-language studies is acknowledged, but its impact is not explored in depth, nor are solutions proposed.
Recommendation: Critically reflect on how this restriction may affect the cultural and methodological diversity of the analysis. Consider including studies in other languages in future versions.
- Conclusions, Limitations and Future Directions
Observation: The conclusion summarises the findings, but lacks depth in the critical analysis of limitations and in outlining future research directions.
Recommendation: Include a more precise discussion of the scope of the findings, methodological biases, and concrete suggestions for future research.
- References
Observation: Updated and relevant. The geographical diversity has been expanded, although Anglo-Saxon literature still predominates.
Recommendation: Consider incorporating studies from other regions and languages to enrich the analysis.
- Final Decision
Rejection. Despite the improvements made, the manuscript still presents significant methodological limitations that affect the quality, validity, and scope of the findings. A thorough revision of the methodological aspects is recommended before the manuscript can be considered for publication.
